# The Interaction between Grains during Columnar-to-Equiaxed Transition in Laser Welding: A Phase-Field Study

**Lingda Xiong [1], Chunming Wang [2], Zhimin Wang [1,\*] and Ping Jiang [1]**

[1]   School of Mechanical Science and Engineering, Huazhong University of Science and Technology, Wuhan 430074, China; d201677183@hust.edu.cn (L.X.); jiangping@hust.edu.cn (P.J.)

[2]   State Key Lab of Material Processing and Die & Mould Technology, Huazhong University of Science and Technology, Wuhan 430074, China; cmwang@hust.edu.cn

\*   Correspondence: d201677150@hust.edu.cn; Tel.: +86-139-1079-2774

**Abstract:** A phase-field model was applied to study CET (columnar-to-equiaxed transition) during laser welding of an Al-Cu model alloy. A parametric study was performed to investigate the effects of nucleation undercooling for the equiaxed grains, nucleation density and location of the first nucleation seed ahead of the columnar front on the microstructure of the fusion zone. The numerical results indicated that nucleation undercooling significantly influenced the occurrence and the time of CET. Nucleation density affected the occurrence of CET and the size of equiaxed grains. The dendrite growth behavior was analyzed to reveal the mechanism of the CET. The interactions between different grains were studied. Once the seeds ahead of the columnar dendrites nucleated and grew, the columnar dendrite tip velocity began to fluctuate around a value. It did not decrease until the columnar dendrite got rather close to the equiaxed grains. The undercooling and solute segregation profile evolutions of the columnar dendrite tip with the CET and without the CET had no significant difference before the CET occurred. Mechanical blocking was the major blocking mechanism for the CET. The equiaxed grains formed first were larger than the equiaxed grains formed later due to the decreasing of undercooling. The size of equiaxed grain decreased from fusion line to center line. The numerical results were basically consistent with the experimental results obtained by laser welding of a 2A12 Al-alloy.

**Keywords:** phase-field model; columnar-to-equiaxed transition; laser welding; interaction

## 1. Introduction

CET (columnar-to-equiaxed transition) occurs when the growth path of columnar grains is blocked by the equiaxed grains form ahead of the columnar front. CET significantly changes the morphology and size of microstructure [1]. It is observed that CET often happens during the solidification process in the molten pool after welding. The microstructure in the fusion zone determines the mechanical properties of welding joints [2]. Therefore, a better understanding of the CET is of great importance to obtain weld joints with high quality.

Numerous experiments have been made to reveal the nucleation mechanism and the factors that affect the CET during solidification process. Gandin et al. conducted a directional solidification experiment with Al–Si alloy. The experimental results showed that nuclei of equiaxed grains may come from heterogeneous nucleation or dendrite arm detachment/fragmentation [3]. Nguyen-Thi et al. observed the dynamic phenomena of the CET during Al-Ni alloy directional solidification by X-ray radiography. They found that the nucleation undercooling of the equiaxed grain depended on the

heterogeneous particle size: larger heterogeneous particles needed lower undercooling than smaller heterogeneous particles [4]. Villafuerte et al. studied the effect of alloy composition on CET in ferritic stainless-steel tungsten inert gas (TIG) welding. It was found that the CET was ascribed to heterogeneous nucleation of ferritic on Ti-rich cuboidal inclusions [5]. Geng et al. adopted an overlap welding procedure to identify the nucleation mechanism in laser welds of aluminum alloys. The result indicated that the nucleation mechanism in laser welding was heterogeneous nucleation, rather than grain detachment and dendrite fragmentation [6].

Modeling has been developed into an important method to study the CET over decades. CET models can be classified into two types: deterministic models and stochastic models. Deterministic models rely on averaged quantities and equations that are solved on a macroscopic scale whereas stochastic models follow the nucleation and growth of each individual grain [7]. Hunt established an analytical model to predict CET in directional solidification [8]. The model has been improved by Kurz et al. to predict the microstructure in welding or other processes involving rapid solidification [9]. Badillo et al. applied the phase-field model to study CET in alloy directional solidification. They found that high pulling velocity and low temperature gradient promote equiaxed growth [1]. Martorano et al. applied a multiphase/multiscale model to study the solutal interaction mechanism for CET in alloy solidification. When the solute rejected from the equiaxed grains was sufficient to dissipate the undercooling at the columnar front, the CET would occur. [10]. Li et al. used the phase-field model to study the CET in alloy solidification. The results indicated that the columnar zone length and the equiaxed grain size increased with the decrease of cooling rate [11]. Viardin et al. conducted the first 3D phase-field simulation of CET and the numerical results were consistent with the experiment [12]. Dong et al. applied a CA-FE (cellular automaton-finite difference) model to study the influence of crystallographic orientation of the columnar dendrites on CET. The simulation indicated that crystallographic orientation had little effect on CET [13]. Biscuola et al. adopted two deterministic model (a model developed by Martorano and a modified Hunt's model) and one stochastic model (CA (cellular automation) model) to study the CET in Al–Si alloy directional solidification. The results indicated that the mechanical blocking occurred in the stochastic model by adopting a blocking fraction of 0.2, rather than 0.49 used in Hunt model [14]. With the development of numerical simulation, several researchers successfully applied multi-phase-field model and CA-FE model to predict CET which occurs in the solidification process in molten pool during metallic welding and additive manufacturing [15–19]. Han et al. used a CA-FE model to predict the CET in the welded pool. They found that the undercooled zone width and the maximum undercooling in front of columnar grains increased with the decrease of the distance to weld center [20]. The above work has greatly improved understanding about the blocking mechanism and influence factors of CET. However, the details of the grain growth and the interactions between different grains during the CET in laser welding have not been studied yet.

In the present work, a phase-field model was used to study the influence of nucleation undercooling, nucleation density and location of first nucleation seed on the CET during laser welding. The dendrite tip behavior and undercooling distribution were analyzed to illustrate the growth of dendrites and equiaxed grains. The blocking mechanisms for CET were discussed. Finally, the experiment was conducted to verify the numerical result.

## 2. Models and Experiments

### 2.1. Macroscopic Model

Zheng et al. proposed a transient condition macroscopic model to obtain the time dependent pulling velocity $V_p$ and thermal gradient $G$ [21]. This model is applied in the present work. The shape of the molten pool is composed of two half ellipsoids, as shown in Figure 1. The temperature at the molten pool edge $T_1$ is equilibrium liquidus temperature. The solidification process occurs in the rear ellipsoid area.

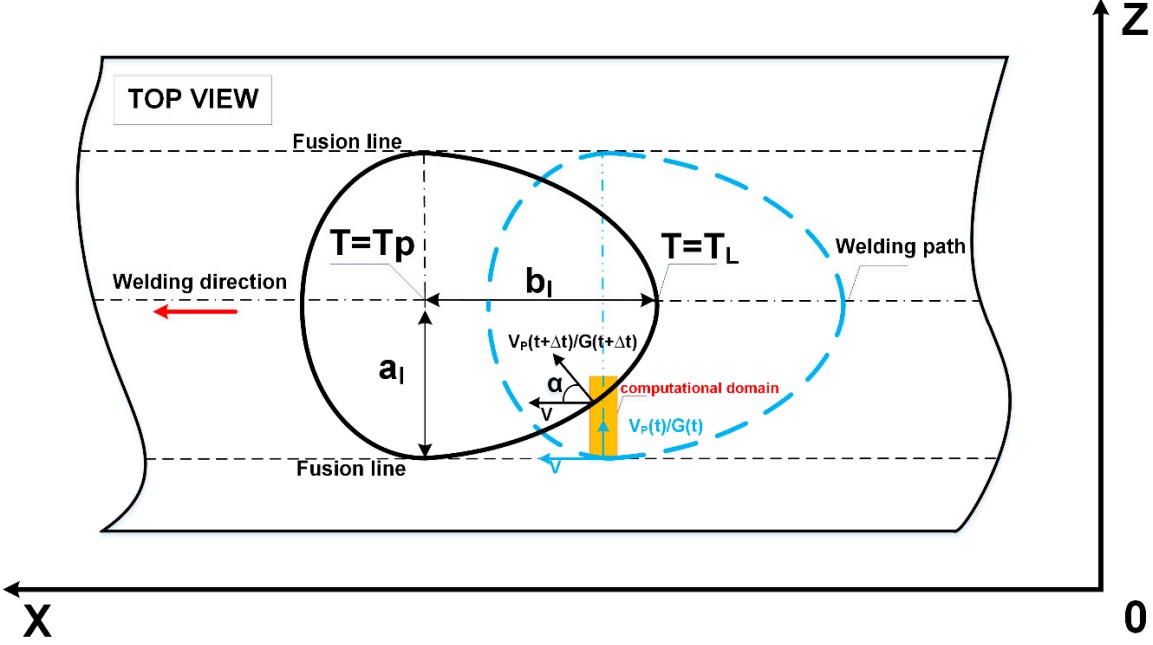

**Figure 1.** Macrograph of the molten pool.

In Figure 1, $a_1$ and $b_1$ are the depth and rear length of the solidification area, respectively. $T_p$ is the highest temperature at the center of molten pool. $V$ is the welding velocity of laser welding. $\alpha$ is the angle between S/L (solid/liquid) interface pulling velocity and welding velocity. The yellow area is the computational domain. $V_p(t)$ is the S/L interface pulling velocity. $G(t)$ is the temperature gradient. Hence, $V_p(t)$ and $G(t)$ in the computational domain can be expressed as:

$$V_p(t) = \frac{a_l V^2 t}{\sqrt{V^2 t^2 (a_l{}^2 - b_l{}^2) + b_l{}^4}}$$ (1)

$$G(t) = \frac{T_p - T_l}{\sqrt{V^2 t^2 + a_l{}^2 \frac{1 - V^2 t^2}{b_l{}^2}}}$$ (2)

*2.2. Phase-Field Modeling*

The phase-field model developed by Zheng was applied to simulate the microstructure evolution during solidification process [21]. The 'frozen temperature approximation' was adopted.

$$T(z,t) = T_0 + G(t)\left(z - \int_0^t V_p(t\prime)dt\prime\right)$$ (3)

where $T_0 = T(z_0,t)$ is a reference temperature, and $G(t)$ is the temperature gradient along pulling direction, the integrand term $\int_0^t V_p(t\prime)dt\prime$ is the pulling distance of S/L interface.

A scalar field $\varphi$ is introduced into the phase-field model to indicate the phase state at every point. In solid phase, $\varphi = 1$. In the liquid phase, $\varphi = -1$. At the S/L interface, $\varphi$ continuously changes from $-1$ to $1$. A dimensionless supersaturation field $U$ is introduced to characterize the solute concentration.

$$U = \frac{1}{1-k}\left(\frac{2kc/c_\infty}{1 - \varphi + k(1 + \varphi)} - 1\right)$$ (4)

where $k$ is equilibrium partition coefficient, $c$ is the solute concentration and $c_\infty$ is the global sample composition.

The governing equations of the phase-field model in two-dimensions can be expressed as the following:

$$\tau_0 a(\widehat{n})^2 [1 - (1-k)\frac{z - \int_0^t V_p(t\prime)d(t\prime)}{l_T}]\frac{\partial\varphi}{\partial t} =$$
$$W^2\vec{\nabla}[a(\widehat{n})^2\vec{\nabla}\varphi] + \varphi - \varphi^3 - \lambda(1-\varphi^2)^2[U + \frac{z - \int_0^t V_p(t\prime)d(t\prime)}{l_T}] \tag{5}$$

$$(\frac{1+k}{2} - \frac{1-k}{2}\varphi)\frac{\partial U}{\partial t} =$$
$$\vec{\nabla}\left(D_L\frac{1-\varphi}{2}\vec{\nabla}U + \frac{1}{2\sqrt{2}}W[1 + (1-k)U]\frac{\partial\varphi}{\partial t}\frac{\vec{\nabla}\varphi}{|\vec{\nabla}\varphi|}\right) + \frac{1}{2}[1 + (1-k)U]\frac{\partial\varphi}{\partial t} \tag{6}$$

where $W$ is the interface width (length scale) and $\tau_0$ is the relaxation time (time scale) respectively, $a(\widehat{n})$ is the anisotropy of the surface tension, $\lambda$ is a coupling constant, $l_T$ is the dimensionless thermal length.

$$W = d_0\lambda/a_1 \tag{7}$$

$$\tau_0 = a_2\lambda W/D_L \tag{8}$$

$$a(\widehat{n}) = 1 + \varepsilon\cos 4\theta \tag{9}$$

$$d_0 = \Gamma/(|m|(1-k)c_l^0) \tag{10}$$

$$l_T = \frac{|m|(1-k)c_l^0}{G(t)} \tag{11}$$

$$c_l^0 = c_\infty/k \tag{12}$$

where $a_1 = 0.88839$ and $a_2 = 0.6267$, $D_L$ is the solute diffusion coefficient in the liquid region, $\theta$ is the angle between the interface normal and pulling direction, $\varepsilon$ is the anisotropy strength, $d_0$ is the capillarity length, m is the liquid slope of alloy, $\Gamma$ is the Gibbs-Thomson coefficient, $c_l^0$ is the equilibrium solute concentration on the liquid side of S/L interface.

*2.3. Computational Details*

The length and width of 2-D computational domain used in the simulations were 3000 $\Delta z$ and 2000 $\Delta x$ respectively. $\Delta x = \Delta z = 0.8W = 2.72 \times 10^{-8}$ m. Therefore, the actual size of computational domain was $8.16 \times 10^{-5}$ m $\times 5.44 \times 10^{-5}$ m. The initial condition is shown in Figure 2. The temperature at S/L interface was 922.6 K. The simulations in the present work were initialized with a thin solid layer (whose width is 100 $\Delta x$) at the left wall. The S/L interface was planar. The reference temperature $T_0$ of S/L interface at the beginning was equilibrium liquidus temperature. The solute was distributed uniformly in the liquid (i.e., solute concentration in the liquid is 4 wt %). The values of parameters involved in phase-field modeling is shown in Table 1.

Wang et al. studied the microstructure evolution during solidification of welding molten pool. They found that the columnar dendrite growth during welding can be divided into four stages: linear growth stage, nonlinear growth stage, competitive growth stage and relatively steady growth stage [22]. In our early stage works, the variation of maximum undercooling ahead of planar S/L interface with time at linear growth stage is shown in Figure 3. We found that the maximum of undercooling ahead of S/L interface during linear growth stage was so large (>17 K) that if the seeds for nucleation were set at the beginning of the simulation, CET would occur before S/L underwent a Mullins–Sekerka instability and columnar dendrites grew when nucleation undercooling was smaller than 17 K. In this circumstance, there would be no columnar dendrites in the fusion zone. This phenomenon was not consistent with what we have observed in the experiment. Therefore, the seeds for nucleation were set in front of the solid layer in the simulations. However, only after entering the relatively steady growth stage (i.e., after $7.0 \times 10^5$ time steps—35% of the overall simulation time), the seeds in the liquid begin to translate into nuclei when the local undercooling (=$T_1$ (C) − T) at the seeds exceeds

the nucleation undercooling, as shown in Figure 4. The seeds for nucleation were distributed in a rectangular array. The horizontal and vertical distances between all adjacent seeds were equal. The radius of heterogeneous nuclei was $2 \Delta x$.

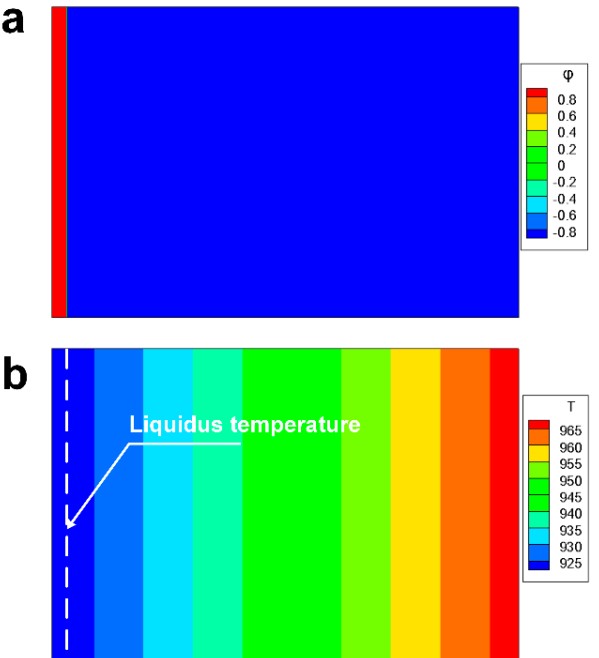

**Figure 2.** The initial condition of simulation: (**a**) the scalar field $\varphi$; (**b**) the temperature distribution.

**Table 1.** The values of parameters involved in phase-field modeling.

| Parameter | Value | Unit |
|---|---|---|
| Coupling parameter: $\lambda$ | 8.0 | - |
| Alloy composition: $c_\infty$ | 4 | wt % |
| Capillary length: $d_0$ | $3.8 \times 10^{-9}$ | m |
| Interface width: $W$ | $3.4 \times 10^{-8}$ | m |
| Partition coefficient: $k$ | 0.14 | - |
| Liquidus slope: $m$ | $-2.6$ | K/wt % |
| Diffusion coefficient: $D_L$ | $3.0 \times 10^{-9}$ | m²/s |
| The anisotropy strength: $\varepsilon$ | 0.01 | - |
| Gibbs–Thomson coefficient: $\Gamma$ | $2.4 \times 10^{-7}$ | K × m |

One of the aims of the present work was to study effects of nucleation undercooling for the equiaxed grains, nucleation density and location of first nucleation seed on CET. Various sets of nucleation undercooling for the equiaxed grains, nucleation density and location of first nucleation seed have been applied in the simulations as shown in Table 2.

Moving-domain technology was applied in the simulations for decreasing simulation time cost. When the temperature of the left wall was higher than 892 K, all fields were shifted by one grid point to the left. The maximum temperature at the right wall was always above the equilibrium liquidus temperature (922.6 K). The upper and lower boundaries were set periodic boundary condition. The zero-Neumann boundary conditions were applied to the other boundaries. The governing equations were discretized by finite difference method. A time step size of $\Delta t = 0.02 \times \tau_0$ $(0.02 < \Delta x^2/4D_L) = 3.88 \times 10^{-8}$ s was chosen to applied in the computation. The evolution of the phase and solute concentration fields were obtained by calculating the governing Equations (5) and (6). The total number of iterations is $2.0 \times 10^7$ (100% of the simulation time), corresponding to actual time 0.0772 s for the solidification process. The self-developed code was

programed by C. Finite difference method with CUDA 9.2 (CUDA Version 9.2.88, NVIDIA Corporation, Santa Clara, CA, USA) parallelization was used to solve the governing equations.

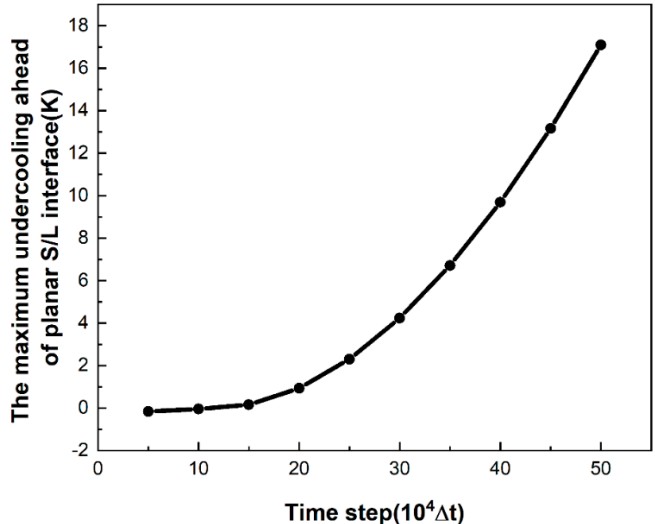

**Figure 3.** The variation of maximum undercooling ahead of planar S/L interface with time.

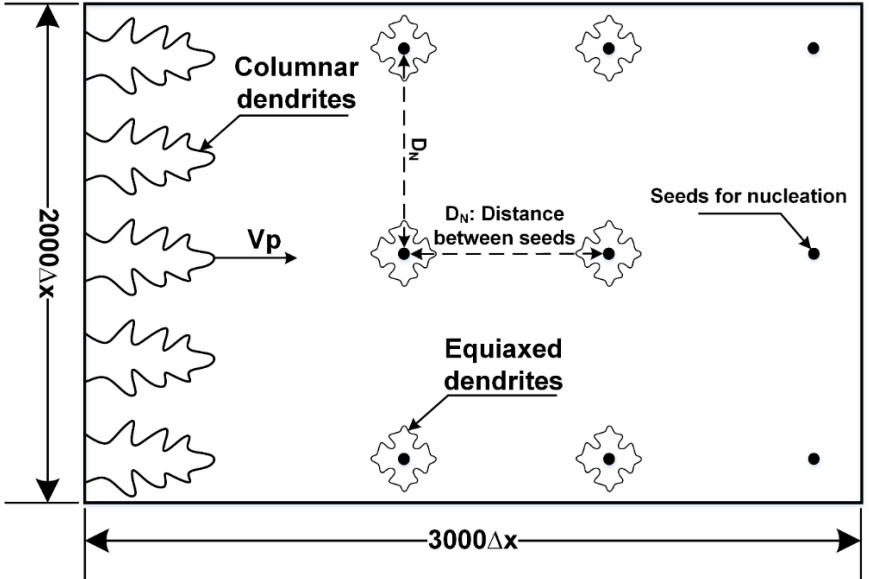

**Figure 4.** The setting of heterogeneous nuclei in the computational domain.

**Table 2.** The growth condition of heterogeneous nuclei.

| Case Number | $U_n$: Nucleation Undercooling (K) | $D_n$: Distance between Nuclei ($\Delta x$) | $X_f$: Location of First Seed at $X$ Coordinate Axis ($\Delta x$) | The Relative Distance between the Initial Planar Front and the First Seeds |
|:---:|:---:|:---:|:---:|:---:|
| 1 | 11 | 200 | 200 | 1578 |
| 2 | 16 | 200 | 200 | 1578 |
| 3 | 21 | 200 | 200 | 1578 |
| 4 | 11 | 400 | 200 | 1578 |
| 5 | 11 | 500 | 200 | 1578 |
| 6 | 11 | 1000 | 200 | 1578 |
| 7 | 11 | 200 | 100 | 1678 |
| 8 | 11 | 200 | 300 | 1878 |

## 2.4. Experiment Design and Material Properties

In order to verify the numerical results, laser welding experiments were conducted. Al-4 wt % Cu alloy 2A12 was used as welded material. Cu is the main component of 2A12 Al-alloy. The content of Cu is 4 wt %. The contents of other chemical elements in 2A12 Al-alloy are very low. The solidification structure of 2A12 Al-alloy can be maintained after the molten pool cool to room temperature. Therefore, the 2A12 Al-alloy can be considered representative for a binary Al-Cu alloy, as used in the simulation. The thickness of the sample was 4 mm. Chemical composition of the material was detected by EDS. It is shown in Table 3. The laser welding equipment was an IPG YLR-4000 fiber laser (IPG Photonics Corporation, Oxford, MA, USA) with a peak power of 4.0 kW and an ABB IRB4400 robot as shown in Figure 5. Pure argon at a flow rate of 2.0 m$^3$/h was used for top surface shielding. The defocusing distance was 0 mm. The laser power was 2.5 kW and the welding velocity was 1.8 m/min. After laser welding, the microstructure at top surface of welded joint was observed by Scanning Electron Microscopy (SEM, Carl Zeiss, Oberkochen, Germany). The size of equiaxed grains at different areas were measured by electron back-scattered diffraction (EBSD, Carl Zeiss, Oberkochen, Germany).

**Table 3.** The chemical composition of 2A12 in wt %.

| Element | Al | Cu | Mg | Mn |
|:---:|:---:|:---:|:---:|:---:|
| wt % | 94.38 | 3.92 | 1.08 | 0.62 |

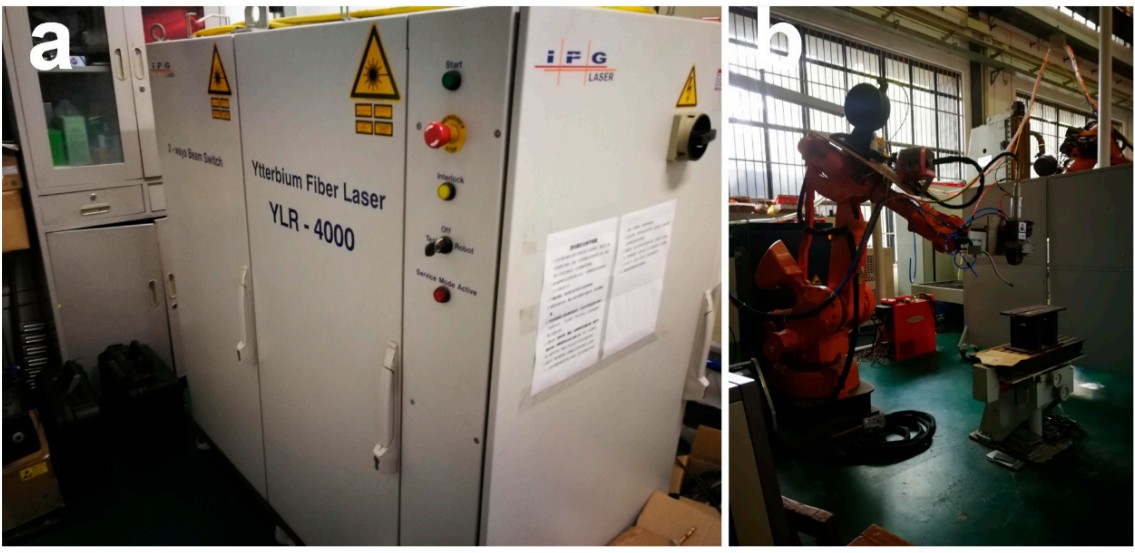

**Figure 5.** Welding devices: (**a**) IPG YLR-4000 fiber laser; (**b**) ABB IRB4400 robot.

The characteristic parameters of the molten pool are listed in Table 4.

**Table 4.** The characteristic parameters of the molten pool.

| Symbol | Value | Unit |
| --- | --- | --- |
| $T_p - T_1$ | 742 | K |
| $a_1$ | $1.2844 \times 10^{-3}$ | m |
| $b_1$ | $2.0825 \times 10^{-3}$ | m |

## 3. Results and Discussion

The microstructure evolution in the molten pool in Case 1 as a typical example is shown in Figure 6. Figure 6a shows the well-developed columnar dendrites just before the CET occurs. The primary dendrite arm spacing was 117.6 $\Delta x$. In Figure 6b, the growth path of columnar dendrites was blocked by the equiaxed grains and columnar dendrites stopped growing. The CET occurred. Figure 6c–h shows the equiaxed grain growth after the CET. It was found that the length of equiaxed grains that nucleated at the end of simulation (in Figure 6h) were smaller than the length of equiaxed grains that formed at the beginning of relatively steady growth stage (in Figure 6c) (i.e., The size of equiaxed grain decreased from fusion line to center line).

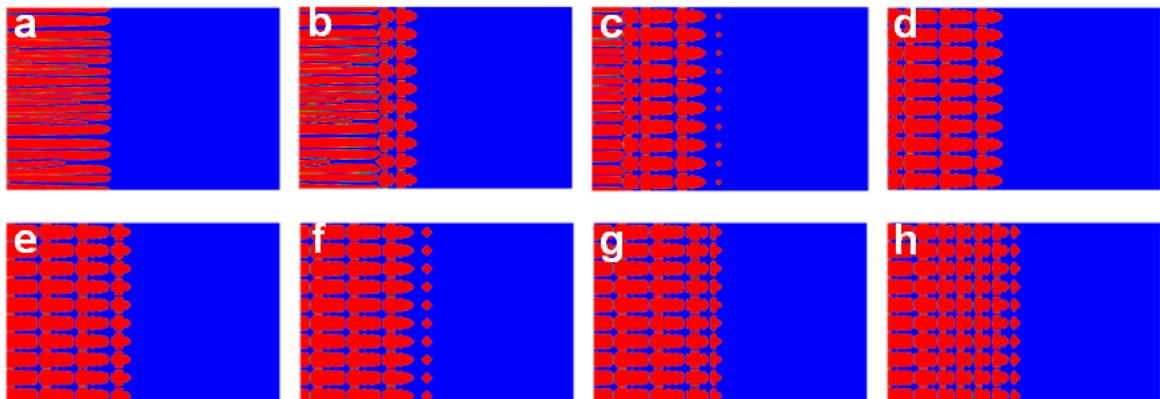

**Figure 6.** The microstructure evolution at different time step: (**a**) $7.0 \times 10^5$ (35.0% of the total simulation time); (**b**) $7.3 \times 10^5$ (36.5% of the total simulation time); (**c**) $7.8 \times 10^5$ (39.0% of the total simulation time); (**d**) $8.1 \times 10^5$ (40.5% of the total simulation time); (**e**) $8.6 \times 10^5$ (43.0% of the total simulation time); (**f**) $9.1 \times 10^5$ (45.5% of the total simulation time); (**g**) $9.6 \times 10^5$ (48.0% of the total simulation time); (**h**) $1.01 \times 10^6$ (50.5% of the total simulation time).

The effects of nucleation undercooling for the equiaxed grains, nucleation density and location of first nucleation seed on the CET were discussed in the following section. In order to illustrate the blocking mechanism for CET and the behaviors of the equiaxed grains after the CET, the variation of dendrite tip velocity, undercooling in front of dendrite tips and the solute segregation at dendrite tips were analyzed to characterize the interactions between columnar grains and equiaxed grains and the interactions between equiaxed grains at neighboring columns.

### 3.1. The Effects of Heterogeneous Nucleation Parameters on Microstructure

Figure 7 shows the microstructures for Case 1–3, where the applied nucleation undercoolings are 11 K, 16 K and 21 K, respectively. It can be seen that when nucleation undercooling was 21 K, no equiaxed grain formed and no CET occurred. When nucleation undercooling was 11 K or 16 K, the CET occurred. However, the equiaxed grains with 16 K nucleation undercooling were bigger than that with 11 K nucleation undercooling. The CET with 16 K nucleation undercooling occurred later

(at $1.32 \times 10^6$ $\Delta t$—66.0% of the total simulation time) than that (at $7.1 \times 10^5$ $\Delta t$—35.5% of the total simulation time) with 11 K nucleation undercooling.



**Figure 7.** Effects of the nucleation undercooling on the CET for $D_n = 200$ $\Delta x$ and $X_f = 200$ $\Delta x$: (**a**) $U_n = 11$ K; (**b**) $U_n = 16$ K; (**c**) $U_n = 21$ K.

According to research by Badillo et al., the maximum undercooling along the symmetry line between columnar dendrites was generally larger than that ahead of dendrite tip in directional solidification [1]. The comparison between the maximum undercooling along the symmetry line between columnar dendrites and the maximum undercooling ahead of dendrite tip when no CET occurs in laser welding is shown in Figure 8. It was found that the maximum undercooling along the symmetry line between columnar dendrites was only a little bit larger than that ahead of dendrite tip. Both undercoolings increased monotonically after entering relatively steady growth stage. Therefore, in the following section, the undercooling ahead of dendrite tip was applied to analyze. However, the undercoolings did not exceed 21 K at the end of simulations when time step = $2.0 \times 10^6$ $\Delta t$. Therefore, CET did not happen with 21 K nucleation undercooling. And the CET with 16 K nucleation undercooling happened later (at $1.32 \times 10^6$ $\Delta t$) than that (at $7.1 \times 10^5$ $\Delta t$) with 11 K nucleation undercooling. The time for the second column of seed to reach nucleation undercooling required more time when nucleation undercooling was 16 K. In this circumstance, the first column of equiaxed grain had more time to grow and become bigger compared to that with 11 K nucleation undercooling.

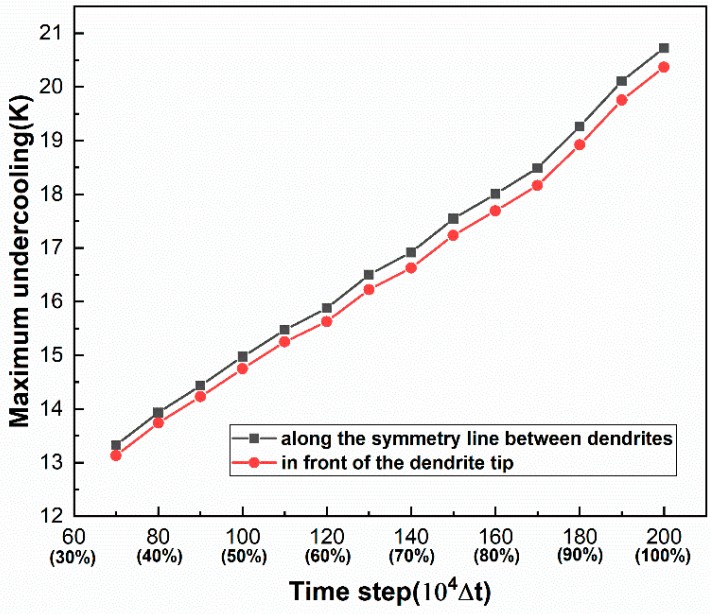

**Figure 8.** The comparison between the maximum undercooling along the symmetry line between columnar dendrites and the maximum undercooling ahead of dendrite tip.

Figure 9 shows the microstructures for Case 1 and 4–6, where $D_n$ (the distance between seeds) are 200 $\Delta x$, 400 $\Delta x$, 500 $\Delta x$ and 1000 $\Delta x$, respectively. It can be seen that when $D_n$ was less than 1000 $\Delta x$,

the CET occurred. The size of equiaxed grains increased with the increasing of distance between seeds. When $D_n$ was 1000 $\Delta x$, no CET occurred and the elongated equiaxed grains and columnar dendrites coexisted in the computational domain. In Figure 9a ($D_n$ = 200 $\Delta x$), the equiaxed grains were elongated while the grains were not elongated in Figure 9b ($D_n$ = 400 $\Delta x$) and Figure 9c ($D_n$ = 500 $\Delta x$). The sidebranchings were well-developed in Figure 9b ($D_n$ = 400 $\Delta x$) and Figure 9c ($D_n$ = 500 $\Delta x$). In Figure 9d, the dendrite arms of elongated equiaxed grains which were vertical to thermal gradient grew along its dendrite direction imposed crystallographic orientation. However, the columnar dendrites next to these dendrite arms always grew so fast that they blocked the growth path of the equiaxed dendrites arms vertical to thermal gradient direction. Therefore, no CET occurred and the final microstructure was a mixed structure (elongated equiaxed grains and columnar dendrites).

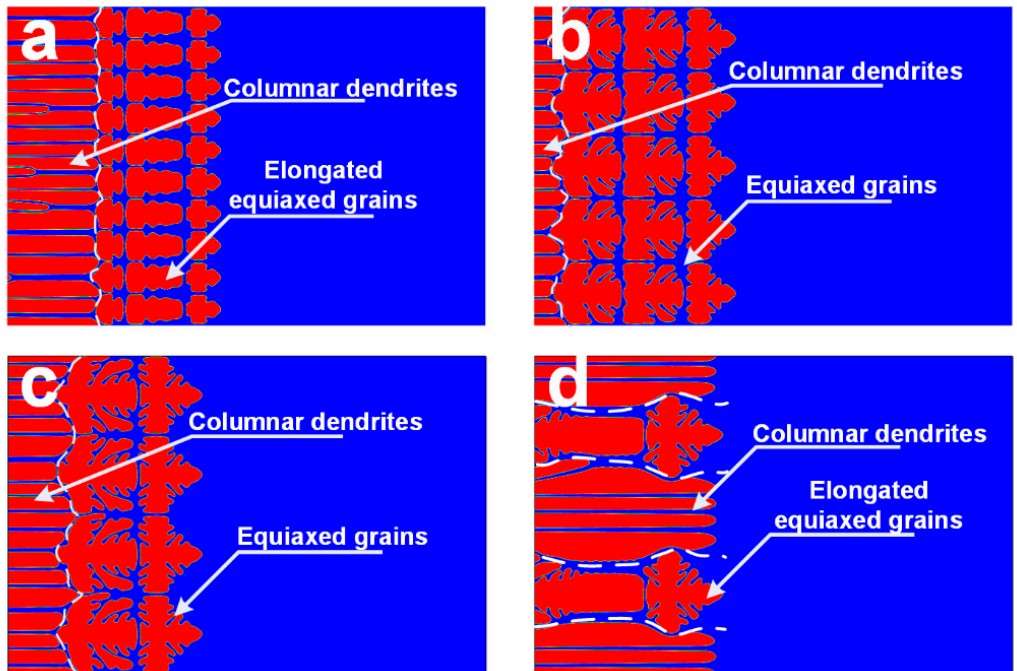

**Figure 9.** Effects of the nucleation density on the CET for $U_n$ = 11 K and $X_f$ = 200 $\Delta x$: (**a**) $D_n$ = 200 $\Delta x$; (**b**) $D_n$ = 400 $\Delta x$; (**c**) $D_n$ = 500 $\Delta x$; (**d**) $D_n$ = 1000 $\Delta x$.

To investigate the effects of the location of first seed on the CET, 100 $\Delta x$, 200 $\Delta x$ and 300 $\Delta x$ were selected with $D_n$ was 200 $\Delta x$. Figure 10 shows the morphologies of equiaxed grains for Case 1, 7 and 8. It was found that the location of first seed significantly influenced the morphology of equiaxed grain microstructure at first column. In Figure 10a, there was only one equiaxed grain. It grew between two columnar dendrites. In Figure 10b, the equiaxed grains at first column grew and blocked the growth path of columnar dendrites. In Figure 10c, the equiaxed grains at first column did not only block the growth path of columnar dendrites but also were elongated. Figure 11 shows the undercooling distribution along the line $x$ = 300 $\Delta x$ where nuclei formed. It was found that in the liquid ahead of the columnar dendrite, with the distance to the dendrite tips increasing, the undercooling increased very slowly at first. Then it increased rapidly to the maximum value. Finally, the undercooling gradually decreased. Figure 11b shows the locations of first emerging nuclei in Case 1, 7 and 8 at $7.0 \times 10^5$ $\Delta t$—35% of the overall simulation time. It was found that in Case 7 (Figure 10a, $D_n$ = 100 $\Delta x$), two columns of nuclei (location: 1122 $\Delta x$ and 1322 $\Delta x$) formed at the same time. In Case 1 (Figure 10b, $D_n$ = 200 $\Delta x$) and 8 (Figure 10c, $D_n$ = 300 $\Delta x$), only one column of nuclei formed since the undercooling at the next column of seeds was below 11 K. The size of equiaxed grains at first column depended on the location where first column of nuclei formed. The closer the distance from the location where first row of nuclei form to point A where the undercooling firstly exceeded 11 K in the liquid, the earlier the

second column of seeds formed. In this circumstance, the time left for equiaxed grains at first column to grow and the space ahead of first column of nuclei decreased. Therefore, the size of the first column of equiaxed grains decreased.

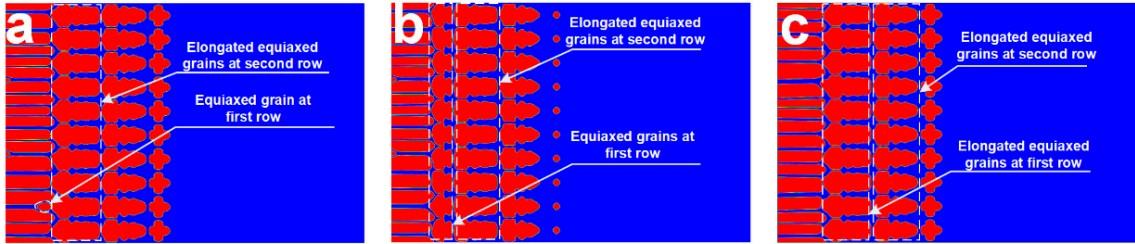

**Figure 10.** Effects of the location of first seed on the CET for $U_n = 11$ K and $D_n = 200\ \Delta x$: (**a**) $X_f = 100\ \Delta x$; (**b**) $X_f = 200\ \Delta x$; (**c**) $X_f = 300\ \Delta x$.

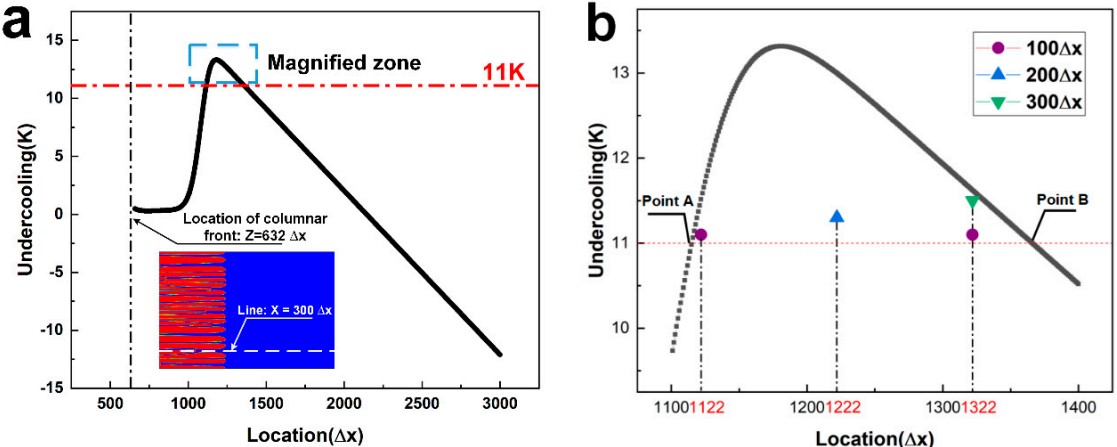

**Figure 11.** (**a**) The undercooling distribution along the line $y = 300\ \Delta x$ at the $7.0 \times 10^5\ \Delta t$; (**b**) The undercooling distribution in magnified zone.

### 3.2. Interaction between Different Grains

To investigate the mechanism of the CET for laser welding, the interaction between different grains were studied. The velocities of columnar dendrite arm and equiaxed grain arm as shown in Figure 12 were selected to study the interaction between grains.

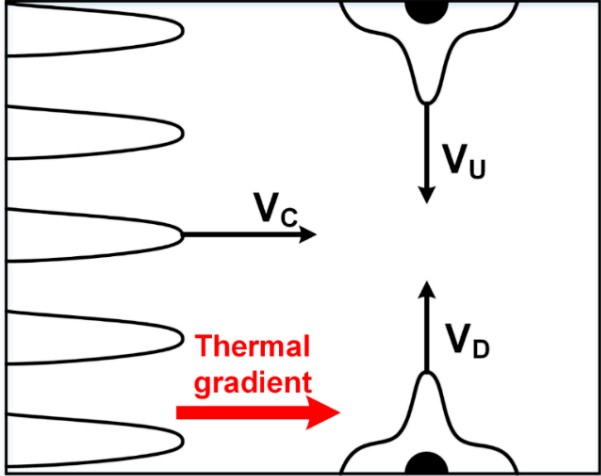

**Figure 12.** The growth of columnar and equiaxed grains.

A dimensionless dendrite tip velocity can be obtained by the following equation:

$$v = -\frac{1}{|\nabla \varphi|} * \frac{\partial \varphi}{\partial t} \tag{13}$$

The actual dendrite tip velocity divided by ($W/\tau_0$) is the dimensionless dendrite tip velocity, $W/\tau_0$ was equal to $1.75 \times 10^{-2}$ m/s.

Without CET, the columnar dendrite tip velocity increased slowly after entering relatively steady growth stage as shown in Figure 13. In Case 1, 4 and 5, the CET occurred when the distance between nuclei increased from 200 $\Delta x$ to 500 $\Delta x$. Figure 14a–c shows the dimensionless velocity and undercooling variations of the columnar dendrite tip until the CET occurs in Case 1, 4 and 5. The selected columnar dendrites analyzed in Figure 14 lied on the line $y = 116$ $\Delta x$ in case 1, line $y = 188$ $\Delta x$ in case 4 and line $y = 242$ $\Delta x$ in case 5, respectively. Figure 14d–f shows the dimensionless velocity and solute segregation variations of the columnar dendrite tip until the CET occurs in Case 1, 4 and 5. As shown in Figure 14a–c, the columnar dendrite tip velocity fluctuated around a value at first. Then the velocity decreased and the columnar dendrite stopped advancing into the liquid after the columnar dendrites got rather close to the equiaxed grains. However, compared to undercooling, the columnar dendrite tip velocity was more related with solute concentration. When the solute segregation at columnar dendrite tip finally increased rapidly, the columnar dendrite tip velocity decreased quickly at the same time. It suggested that the constitutional undercooling played a more important role on the decreasing of columnar dendrite tip velocity. The blue pans in Figure 14d–f show the time step when the solute layers of columnar dendrites and equiaxed grains contacted with each other. It was interesting that the columnar dendrite tip velocity did not decrease until the solute layers of columnar dendrites and equiaxed grains have interacted with each other for some time, but not immediately after the solute layers of these dendrites interact with each other. As mentioned above, the columnar dendrite tip velocity increased continuously after entering relatively steady growth stage without CET. Therefore, the contact of solute layers of columnar dendrites and equiaxed grains firstly prevented the columnar dendrite tip velocity from increasing. The columnar dendrite tip velocity fluctuated around a value. After the solute in front of the columnar dendrite tip accumulated for some time, the columnar dendrite tip velocity started to decrease.

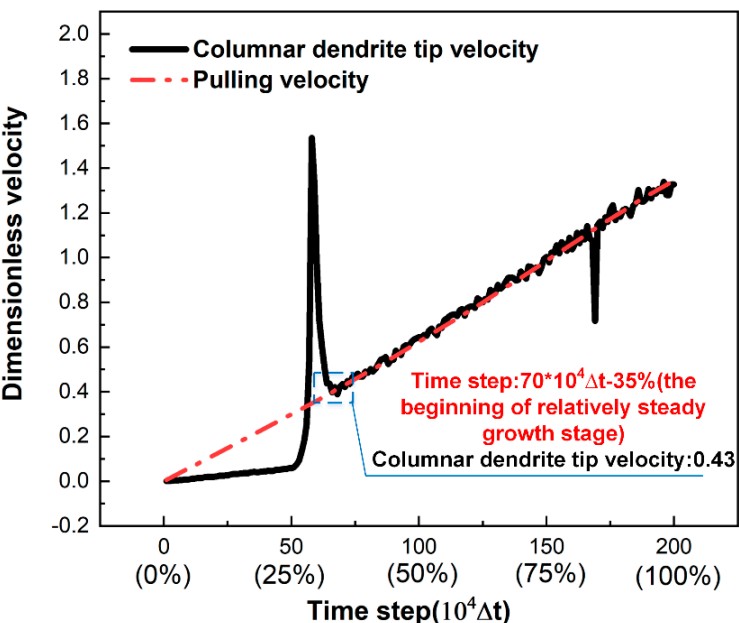

**Figure 13.** Columnar dendrite tip velocity evolution without CET.

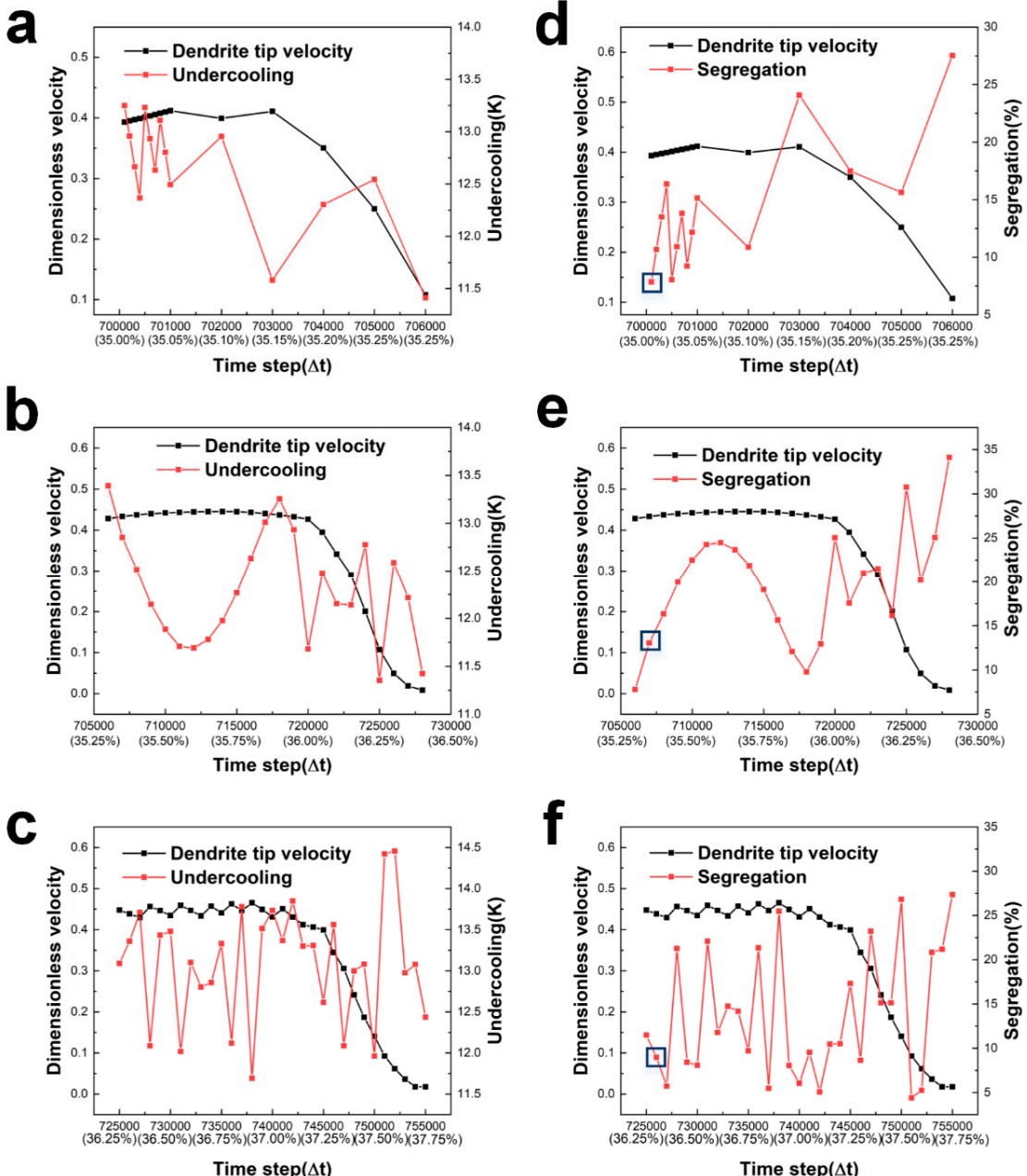

**Figure 14.** The velocity and undercooling variations of columnar dendrite tip with time for $U_n$ = 11 K and $X_f$ = 200 $\Delta x$: (**a**) $D_n$ = 200 $\Delta x$; (**b**) $D_n$ = 400 $\Delta x$; (**c**) $D_n$ = 500 $\Delta x$. The velocity and solute segregation variations of columnar dendrite tip with time for $U_n$ = 11 K and $X_f$ = 200 $\Delta x$: (**d**) $D_n$ = 200 $\Delta x$; (**e**) $D_n$ = 400 $\Delta x$; (**f**) $D_n$ = 500 $\Delta x$.

Figure 15 shows the dimensionless velocity, undercooling and solute segregation variations of the dendrite arm tip of equiaxed grains vertical to thermal gradient until the CET occurs in Case 1, 4 and 5. It was found that the dendrite tip velocity was very high soon after the seed nucleated and grew. Then the velocity decreased rapidly to around the value 0.4. After that, the velocity fluctuated around 0.4. Finally, the dendrite arms met together and the tip velocity decreased to 0. The undercooling of dendrite arm tip mainly kept increasing before the dendrite arms got too close. It began to decrease with the final increase of the solute segregation at the dendrite arm tip. The solute segregation at the dendrite arm tip kept increasing when the equiaxed grain dendrite arm grew.

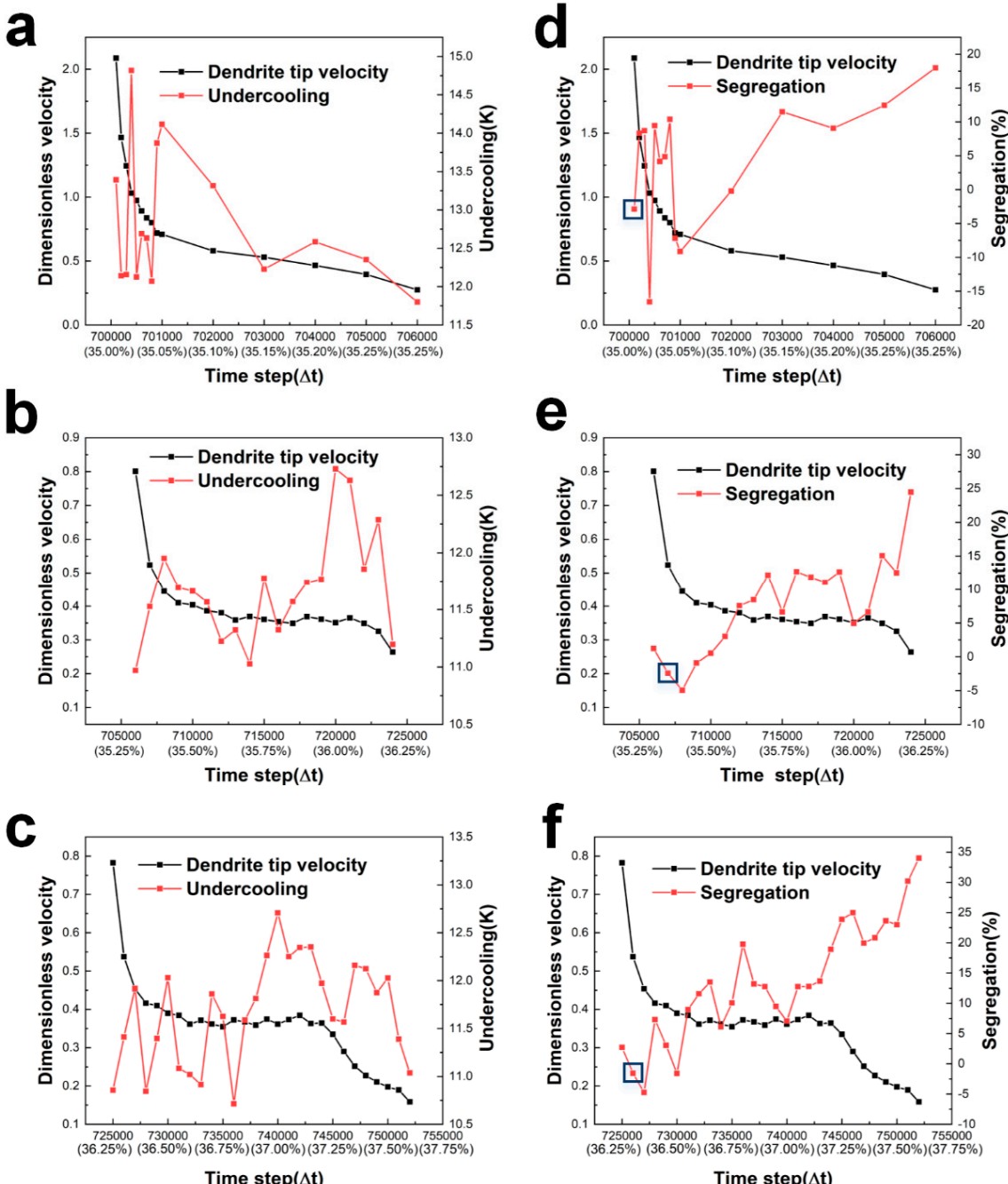

**Figure 15.** The dimensionless velocity and undercooling variations of equiaxed dendrite tip with time for $U_n$ = 11 K and $X_f$ = 200 $\Delta x$: (**a**) $D_n$ = 200 $\Delta x$; (**b**) $D_n$ = 400 $\Delta x$; (**c**) $D_n$ = 500 $\Delta x$. The velocity and solute segregation variations of columnar dendrite tip with time for $U_n$ = 11 K and $X_f$ = 200 $\Delta x$: (**d**) $D_n$ = 200 $\Delta x$; (**e**) $D_n$ = 400 $\Delta x$; (**f**) $D_n$ = 500 $\Delta x$.

Figure 16 shows the comparison of dimensionless columnar dendrite tip velocity, undercooling and solute segregation between Case 5 ($U_n$ = 11 K, $X_f$ = 200 $\Delta x$ and $D_n$ = 500 $\Delta x$) in which the CET occurs and Case 6 ($U_n$ = 11 K, $X_f$ = 200 $\Delta x$ and $D_n$ = 1000 $\Delta x$) in which no CET occurs. It was found that no significant difference existed between the undercoolings and solute segregations of Case 5 and 6. However, the CET occurred when the columnar grains and equiaxed grains got rather close to each other. No space was available for columnar dendrite growth. Therefore, it was concluded that the CET was mainly caused by mechanical blocking but not thermal nor solute blocking.

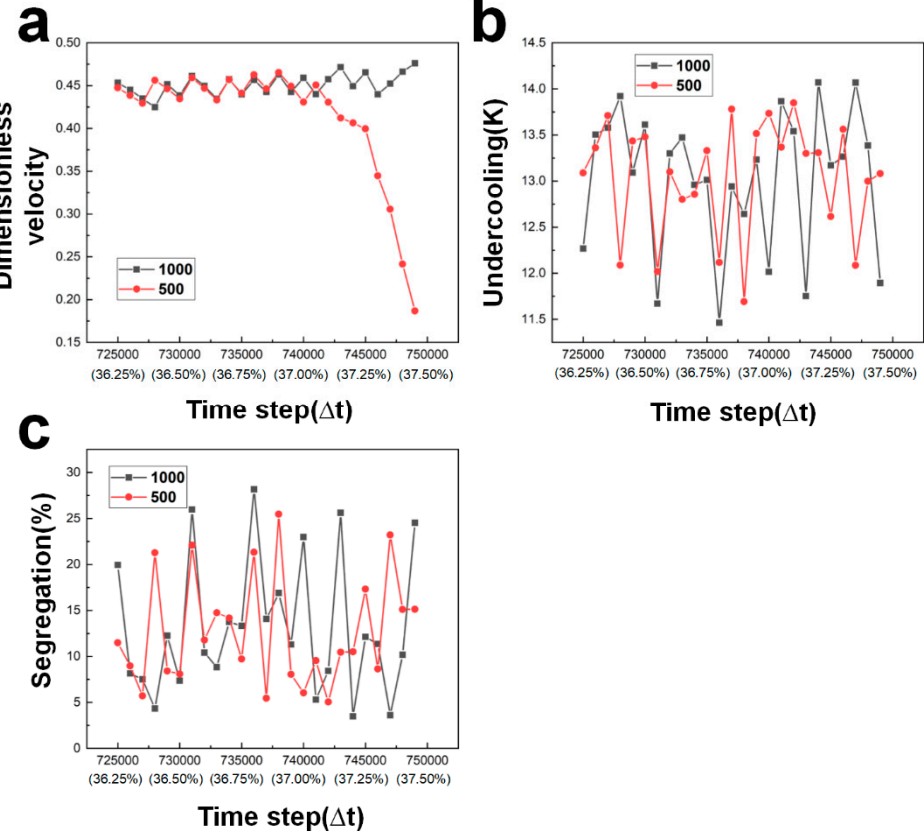

**Figure 16.** The comparisons of: (**a**) columnar dendrite tip velocity; (**b**) undercooling of dendrite tip; (**c**) solute segregation between case 5 and 6.

In Case 1 ($U_n$ = 11 K, $X_f$ = 200 $\Delta x$ and $D_n$ = 200 $\Delta x$), equiaxed grains that nucleated later were smaller than equiaxed grains that nucleated at the beginning of relatively steady growth stage after the CET occurred. This phenomenon indicated that the growing behaviors of equiaxed grains nucleating at different times were different. In order to illustrate this phenomenon, the microstructure evolution after CET was analyzed.

Until the first nucleation occurred at $7.0 \times 10^5 \Delta t$, the frontier of columnar dendrite had accumulated amount of solute. To ensure the reliability of the analysis, the second column of nuclei were used to study the growth behaviors of equiaxed-grain dendrite arm parallel to thermal gradient as shown in Figure 17 to avoid the effect of solute layer ahead of columnar dendrites.

Figure 18 shows the evolution of dimensionless dendrite tip velocity, undercooling and solute segregation. It was found that the solute segregation at dendrite tip was small (less than 1%). The overall solute segregation decreased. The decrease of solute concentration increased the undercooling of dendrite tip. Therefore, the overall undercooling increased. These factors promoted the increasing of dendrite tip velocity. However, the dendrite tip velocity was very high at the beginning. Then it decreased very fast to 0.25 and fluctuated. Finally, the dendrite tip was stopped by the third column of nuclei.

To illustrate the reason for change of equiaxed grain size with time, it was essential to study the undercooling distribution ahead of the equiaxed grain in the liquid and the undercooling evolution with time. As shown in Figure 19a, the undercooling in liquid increased at first, then decreased continuously with the increasing of distance to the equiaxed grains. Furthermore, the value of the maximum undercooling and its distance to the equiaxed grains increased with time as shown in Figure 19b.

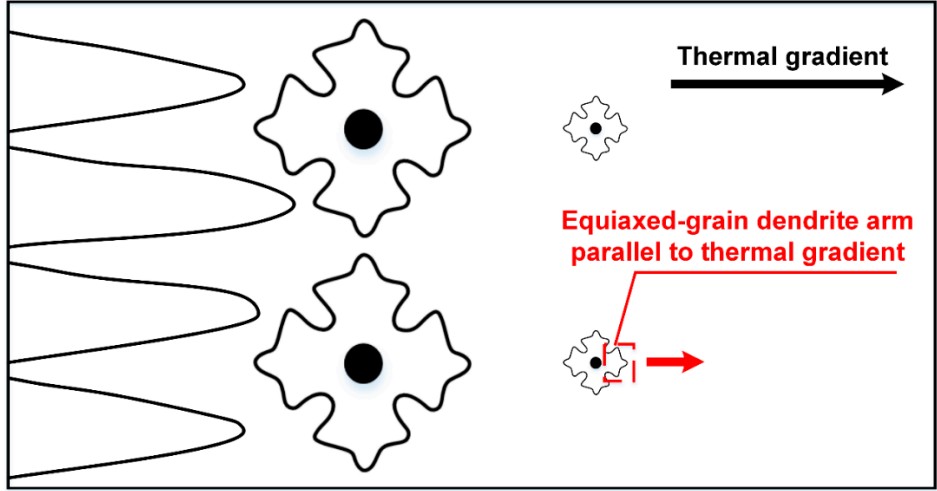

**Figure 17.** The diagram of second row of nuclei.

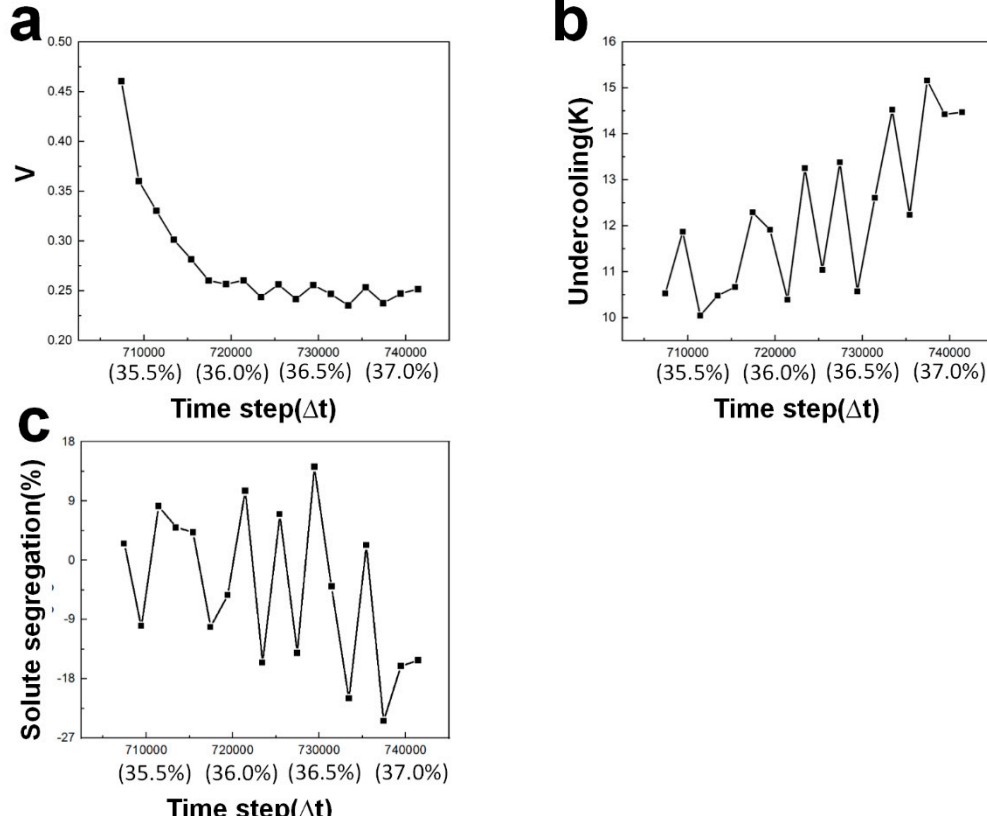

**Figure 18.** The evolution of (**a**) dendrite tip velocity; (**b**) undercooling; (**c**) segregation.

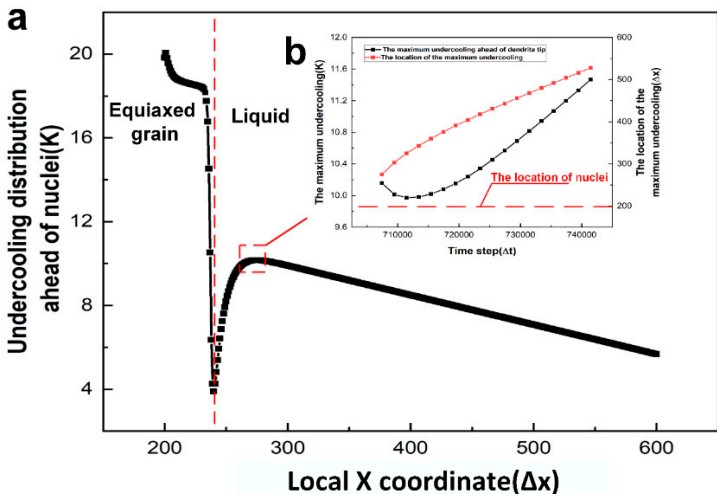

**Figure 19.** (**a**) The undercooling distribution ahead of nuclei; (**b**) The value and location variations of maximum undercooling in the liquid in Figure 17a with time.

The time interval of each column of equiaxed-grain formation (i.e., the growing time for equiaxed grains in the front column) is shown in Figure 20. Except for the first column of equiaxed-grain formation which were affected by columnar dendrite growth, the growing time for 2nd–7th columns of equiaxed grains was significantly longer than the following columns of equiaxed grain. Therefore, the 2nd–7th columns of equiaxed grains were elongated. The undercooling distributions ahead of nucleated seeds when the nuclei were formed at 705,430 $\Delta t$ (2nd column) and 1,990,720 $\Delta t$ (134th column) are shown in Figure 20b. At the location of nuclei, the undercooling ahead of nuclei in the liquid at 705,430 $\Delta t$ (2nd) and 1,990,720 $\Delta t$ (last column) were calculated by following expression.

$$U_n \ = \ T_0 - mc_n - (T_1 + GD_n) \tag{14}$$

$$U \ = \ T_0 - mc - (T_1 + GD) \tag{15}$$

where $U$ is undercooling of selected point ahead of the nuclei, $U_n$ is the undercooling at seed position just before nucleation i.e., 11 K, $T_0$ is the melting point of pure Al, $T_1$ is the temperature of reference point, $D$ is the distance from selected point to reference point, $D_n$ is the distance from nuclei to reference point. Therefore, the relationship between $U$ and $U_n$ is shown as Equation (16) (Equations (14) and (15)):

$$U \ = \ U_0 + m \times (C_0 - C) - G \times D_s \tag{16}$$

where $D_s$ is the distance from the selected point to nuclei.

When a nucleus formed, the solute around the nucleus did not diffuse for a long distance. Therefore, it can be assumed that the solute concentration ahead of the nucleus in the liquid was not affected by the nucleation. In this condition, the undercooling of selected point was mainly affected by the temperature gradient. As shown in Figure 20c, the temperature gradient decreased with time. Therefore, the undercooling ahead of nuclei in the liquid at 705,430 $\Delta t$ (the 2nd column) was smaller than that at 1,990,720 $\Delta t$ (the 134th column). The undercooling at the position where the 3rd column of nuclei set at 705,430 $\Delta t$ was lower than that at the position where the 135th column of nuclei set at 1,990,720 $\Delta t$. The 3rd column of nuclei needed more time to form than the 135th column of nuclei. As a result, the 2nd–7th columns of equiaxed grains had more time to grow. Therefore, the size of equiaxed grain decreased from fusion line to center line.

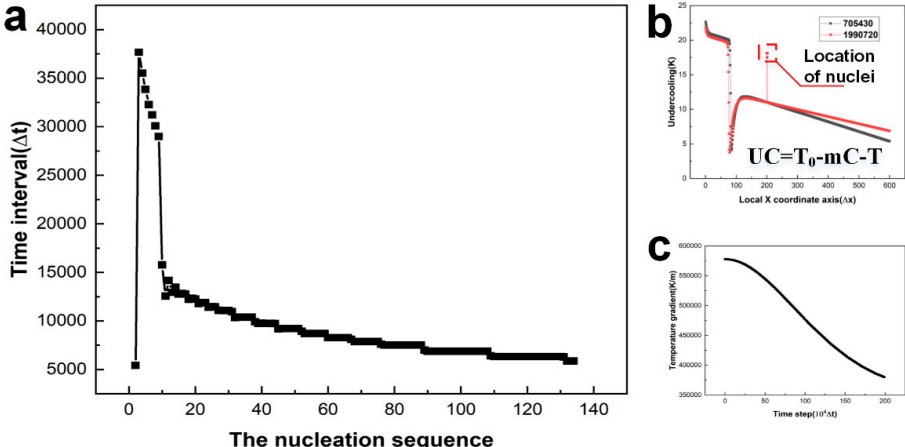

**Figure 20.** (**a**) The time interval of each row of equiaxed-grain formation; (**b**) undercooling distribution when seeds nucleate; (**c**) temperature gradient variation with time.

### 3.3. Experimental Results

Figure 21 shows the microstructure at the top surface of the fusion zone. It was found that the fusion zone was consisted of 3 zones: (a) columnar grain zone; (b) mixed microstructure zone; (c) Equiaxed grain zone. The grain size of equiaxed grains was measured. As shown in Figure 22, it was found that the size of equiaxed grain decreased from fusion line to center line, corresponding to the numerical result.

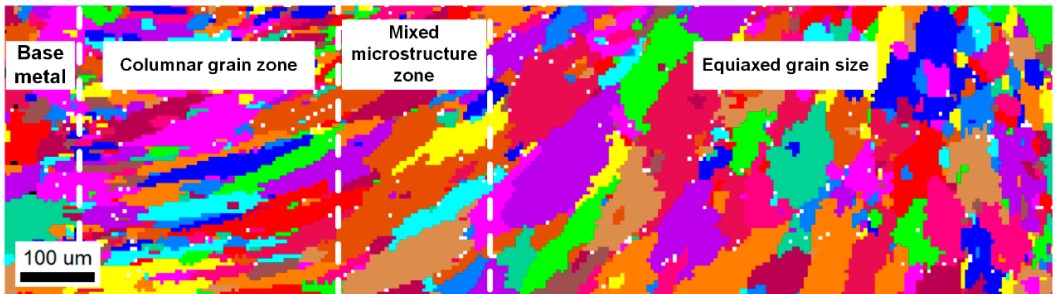

**Figure 21.** The microstructure of fusion zone.

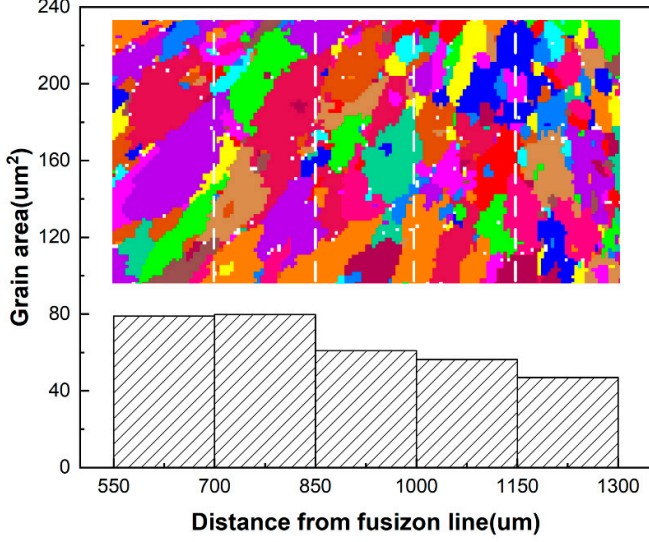

**Figure 22.** The size distribution of equiaxed grain in the fusion zone.

## 4. Conclusions

The CET during laser welding was investigated using a 2-D phase-field model. Experiments were conducted to verify the numerical result. The conclusion are as follows:

(1) Nucleation undercooling significantly influenced the occurrence and the time of CET. Nucleation density affected the occurrence of CET and the size of equiaxed grains.

(2) The contact of solute layers of columnar dendrites and equiaxed grains firstly prevented the columnar dendrite tip velocity from increasing. The solute blocking decreased the dendrite tip velocity of columnar grains before the CET happened. The mechanical blocking was the major mechanism for the CET.

(3) The decrease of the temperature gradient during solidification of the laser welding molten pool led to the decreasing of size of equiaxed grain from the fusion line to the center line.

It should be noted, however, that the microstructure evolution in real 3-D space during welding is more complex than a 2-D phase-field numerical simulation can predict. Nonetheless, 2D phase-field models have been applied to simulate CET during solidification for decades [1,11,18]. The results have shown a reasonable quantitative agreement between simulations and the analytical model for CET. However, a 2D model assumes an in-depth homogeneity of the system, a condition that prohibits modeling of fully-realistic 3D microstructural features. In a 2D model, Pe or the direction of crystalline orientations are limited to a smaller range than in a 3D model [23]. The solute diffusion during solidification can be completely blocked by the dendrite arms in 2D simulations, while it can bypass the 3D arms [24]. Interactions between crystals in the solidification process in real 3D physical space are more likely to appear between different layers. This effect cannot be simulated by a 2D model [25,26]. When the fluid flow is taken into consideration, the differences in the flow pattern in front of dendrite tips lead to the differences of dendrite growth behavior between 2D and 3D simulations [27]. All the above differences produce different distributions of solute, growth kinetics, and dendrite morphology between a 2D model and the 3D reality, limiting the accuracy of a 2D model for real 3D dendrite growth behaviors [23,27–30].

In principle it would have been interesting to compare our simulation results with Hunt's model. In Hunt's model, it is assumed that the columnar front is blocked, causing the CET, when the equiaxed grain fraction at the front is equal to or larger than a predetermined blocking fraction. Hunt's model can be applied for CET prediction in the solidification of laser welding molten pool if the temperature distribution and variation are known. However, in our present work, we focused on the interaction between the columnar grains and equiaxed grains and the computational domain was very small (81.6 × 54.4 um) compared to the whole columnar grain front in Hunt's model. The computational domain in our work can thus not represent the whole columnar grain front in Hunt's model. A comparison between our simulations and Hunt's model therefore lacks comparability, and, any comparison with Hunt's model has therefore not been in the present work. However, in the future, we plan to predict the whole microstructure of the laser welding fusion zone by phase-field modeling using multi-graphics processing units (GPUs) parallel computing. In this case, it will be very useful and meaningful to compare the results of our future work with Hunt's model.

**Author Contributions:** Conceptualization, Z.W.; methodology, L.X.; software, Z.W.; validation, L.X.; formal analysis, L.X.; investigation, L.X.; resources, P.J.; data curation, L.X.; writing—original draft preparation, L.X.; writing—review and editing, Z.W.; visualization, L.X.; supervision, Z.W.; project administration, C.W.; funding acquisition, C.W. All authors have read and agreed to the published version of the manuscript.

**Funding:** This research was funded by the National Natural Science Foundation of China, grant number 51861165202, the National Natural Science Foundation of China, grant number 51705173 and Science and Technology Planning Project of Guangdong Province, grant number 2017B090913001. The APC was funded by the National Natural Science Foundation of China, grant number 51705173.

**Acknowledgments:** The authors are grateful to the Analysis and Test Center of HUST (Huazhong University of Science and Technology) and the State Key Laboratory of Material Processing and Die & Mould Technology of HUST, for their friendly cooperation.

**Conflicts of Interest:** The authors declare no conflict of interest.

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
