# Peer review of "The Interaction between Grains during Columnar-to-Equiaxed Transition in Laser Welding: A Phase-Field Study"

_metals, doi:10.3390/met10121647_

Round 1

Reviewer 1 Report

The authors have studied factors influencing the Columnar-to-Equiaxed transition in laser welding through parametric study employing phase-field approach. Effect of nucleation undercooling and density is investigated by considering three temperatures and  four different distance(s) between the nuclei, respectively. Moreover, influence of the position is studied under two settings. Experimental observations are reported to corroborate the outcomes of numerical investigations. The manuscript is well-written and apparently, unravels novel findings which might deepen the current understanding of the microstructural evolution accompanying welding. Therefore, the reviewer recommends the publication of the manuscript after few minor revisions which are mentioned below.

  1. Even though the expansion of CET is mentioned in the abstract, the reviewer encourages it to be included in the body of the manuscript as well, given its importance and usage. And indeed, the first line of introduction begins with it. 
  2. Expansions of the abbreviations like MGB and CA model cannot be explicitly found in the manuscript. 
  3. Though S/L interface is expanded in line 99 of page 3, it is first introduced in line 89 of page 2. Thus, the expansion should be mentioned earlier. 
  4. Verify the usage of refs 21, 22, and 23 in the manuscript. In its present form, it seems incorrect.
  5. In Eqn.3, the integrand term with $t'$ can be briefly explained for the understanding of the reader.
  6. Values of parameters involved in phase-field modelling, including coupling constant and capillary length, along with its units cannot be found in the manuscript. It would be absolutely helpful if they are tabulated separately.
  7. In line 121 of page 4, please include the reference for the authors' previous works.
  8. Check the value of interface width in line 142 of page 5. 
  9. In line 174 of page 7, replace 'stoppd' with 'stopped'.
  10. While the efforts of the authors to substantiate their results with experimental studies is hugely appreciated, according to the reviewer, it is bit disappointing to see the ensuing discussion relating the observed and simulated results. In other words, only one aspect of the simulated result, i.e size of the equiaxed grains, is discussed in relation to the experimental observations. 

Reviewer 2 Report

The interaction between grains during Columnar-to-Equiaxed transition in laser welding: A phase-field study

The authors compare an established 2D phase-field method for transient directional solidification with inclusion of seeding and nucleation to study effects that influence CET in an Al-4wt%Cu model alloy during laser welding. While such kind of study has been performed in literature for moderate thermal gradients and pulling velocities, it is applied here for laser welding. The article is well structured, but there are some important issues not captured/discussed.

  • The choice of a 2D phase-field model for a 3D experimental situation is not discussed. As such, the comparison to experiment remains qualitative
  • The effect of different alloys studied: Al-Cu binary model alloy in simulations and a more complex Al alloy in the experiments is not discussed
  • As major outcome the author’s claim “mechanical blocking” to be the dominant mechanism for grain-grain interaction. A comparison to Hunt’s model for this situation would be straightforward and extend the applicability of the results to other situations.
  • The effects of rapid solidification in laser welding for the choice of the model have not been discussed.

As a sidemark: for a reviewer it is very tedious to recall the use of units for numbers, the limitation of digits in numerical values, the correct use of citations, etc…It is also not a big deal to use a word processor in English language to avoid typo’s, grammatical errors, etc…

Abstract:

Line-12: Please highlight the differences between modelling and experiments here. “…Al-Cu model alloy”

Line-14: The location of the first seeds should be related to the initial columnar front, which seems more physical. “location of first nucleation seeds ahead of the columnar front”

Line-19: Better use: “…began to fluctuate around a value

Line-20: “It did not decrease until the columnar dendrite got rather close…”

Line-20: “The undercooling and solute segregation profile evolutions…”

Line-24: “The numerical results were basically consistent with the experimental results obtained by laser welding of a 2A12 Al-alloy.”

Introduction:

Line-29: “CET occurs when the growthing path of columnar grains is blocked by the equiaxed grains that form…”

Line-32: “The microstructure in the fusion zone determines the mechanical properties of welding joints[2].”

Line-37: Basically there is no indication in the paper from Gandin et al. [3], that breakdown of the columnar front or fragmentation is active in the specific experiments in [3].

Line-38: The author is Nguyen-Thi, not Thi.

General: Please change et al to et al. throughout the whole paper

Line-53: Please cite the main author..Badillo et al.

Line-56: “The results showed that CET occurs abruptly, with nucleation of equiaxed grains only took place over a small distance ahead of the columnar front[10].” This statement is not correct, in [10] also mixed zones exist, depending on the simulations parameters.

General: The authors conclude that “mechanical blocking” is the dominant mechanism for CET in their simulations. A comparison to Hunt’s model would be of great benefit. Especially, because the author’s summarized other author’s using this approach (i.e. Biscuola et al.) or even Badillo et al. have used this approach. The main outcome could be a discussion about the applicability of Hunt’s model to laser welding.

General: The author’s should clearly motivate the use of a 2D simulation tool for a 3D solidification problem.

Models and Experiments:

General: Please include a consistent coordinate system in Figure 1. Here “z” is included, while in the simulations finite differences are calculated in “x,y” directions. The graphical representation of the calculation domain is not consistent or confusing. The dendrites are supposed to grow along their 3 crystallographic orientations with one of them along the pulling direction. That one is along the “alpha” arrow in Figure 1, which would mean, that the dendrites would grow inclined in the calculation domain.

Line-114-116: Please give the domain size also in units of one of the ellipsoidal parameters.

Line 118: wt.-%

Line 117: The solute is distributed…

Line 119: stages

Line 127: Please give the times in percentage of simulation time also

Line 136 and Table 1: Although clear from Fig. 2, a comment on the relative distance between the initial planar front and the first seeds would be valuable (and can be included as comment in Table 1.

Line 140: Please include a sketch with the initial temperature distribution, including Liquidus-temperature.

Line-142: We guess, that the power is -8 and not 8?

General: Please include the different length scales somewhere in the article: diffusion length, interface thickness, capillary length, numerical resolution.

Line 144: unit of time-step?

Line 147: Please scale simulation times with the maximum simulation time, see comment above.

Line 153: Thickness of sample

Results and discussion:

Line 174: stopped

Line 174-177: The described effect is not very obvious in the selected graphs

Legend Figure 4: Scaling of time-steps, see above

Line 187: The effects of heterogeneous nucleation on growth conditions

General: Please use consistently a blank or no blank between numbers and units in the article

Line 192-193: Scaling of time-steps, see above

Line 197: Badillo et al.

Legend Figure 6: Scaling of time-steps, see above

Line 222: along its dendrite direction imposed crystallographic orientation

Line 230: were selected with Dn is 200

Line 232 (and later in the article): I guess the author’s meant “column” and not “row”. Column is vertical in the figures.

Line 235: “Fig.9 showed the undercooling distribution along the horizontal line y=300 â–³x where nuclei formed”. Please indicate the line in the figures and the time-step should be given.

Figure 9: The figure is completely unclear. Is it meaningful to calculate an undercooling in the solid phase? The position of the columnar front should be indicated in the graph.

Line 238: decreased

Line 242: The size of equiaxed

Line 255-260: The figure is misleading, because it explains blocking only in case of “mechanical blocking”. Columnar dendrites may also be blocked by solutal interaction. Although the authors propose mechanical blocking to be the main mechanism, this result should not be used for a general explanation of the blocking mechanism, as in Fig. 10.

Line 263-265: repetition, not needed.

Fig. 11: No units at the vertical scale!?! No units for the numbers in Figure!?! Too much digits for the number. Time step scaling. There should be a reference, line the welding velocity and the pulling-velocity in the same graph! It remains unclear, how Fig. 11 contributes to the discussion of interactions mechanisms. What is the message?

Line 266: which columnar tip?

Fig. 12: No units at the vertical scale!?! Time step scaling. It is unclear, which incidents occur on the horizontal line. Too small size of text.

Is stop the review here, because this information is important for further reading…
